

# Larval growth rate is not a major determinant of adult wing shape and eyespot size in the seasonally polyphenic butterfly *Melanitis leda*

Freerk Molleman[1], M. Elizabeth Moore[2], Sridhar Halali[3], Ullasa Kodandaramaiah[4], Dheeraj Halali[5], Erik van Bergen[6], Paul M. Brakefield[5] and Vicencio Oostra[7]

[1] Department of Systematic Zoology, Adam Mickiewicz University of Poznan, Poznań, Poland
[2] Emerging Pests and Pathogens Research Unit, Robert W. Holley Center for Agriculture and Health, Ithaca, New York, United States
[3] Department of Biology, Lund University, Lund, Sweden
[4] IISER-TVM Centre for Research and Education in Ecology and Evolution (ICREEE), Indian Institute of Science Education and Research Thiruvananthapuram, India, Vithura, Kerala, India
[5] Department of Zoology, University of Cambridge, Cambridge, United Kingdom
[6] Center for Ecology, Evolution and Environmental Changes (cE3c) & Global Change and Sustainability Institute (CHANGE), Faculty of Sciences, University of Lisbon (FCUL), Lisbon, Portugal
[7] School of Biological and Behavioural Sciences, Queen Mary University London, London, United Kingdom

Corresponding author
Freerk Molleman,
fremol@amu.edu.pl

## ABSTRACT

**Background:** Insects often show adaptive phenotypic plasticity where environmental cues during early stages are used to produce a phenotype that matches the environment experienced by adults. Many tropical satyrine butterflies (Nymphalidae: Satyrinae) are seasonally polyphenic and produce distinct wet- and dry-season form adults, providing tight environment-phenotype matching in seasonal environments. In studied Mycalesina butterflies, dry-season forms can be induced in the laboratory by growing larvae at low temperatures or on poor food quality. Since both these factors also tend to reduce larval growth rate, larval growth rate may be an internal cue that translates the environmental cues into the expression of phenotypes. If this is the case, we predict that slower-growing larvae would be more likely to develop a dry-season phenotype.

**Methods:** We performed the first experimental study on seasonal polyphenism of a butterfly in the tribe Melanitini. We measured both larval growth rate and adult phenotype (eyespot size and wing shape) of common evening brown butterflies (*Melanitis leda*), reared at various temperatures and on various host-plant species. We constructed provisional reaction norms, and tested the hypothesis that growth rate mediates between external cues and adult phenotype.

**Results:** Reaction norms were similar to those found in Mycalesina butterflies. We found that both among and within treatments, larvae with lower growth rates (low temperature, particular host plants) were more likely to develop dry-season phenotypes (small eyespots, falcate wing tips). However, among temperature treatments, similar growth rates could lead to very different wing phenotypes, and within treatments the relationships were weak. Moreover, males and females

responded differently, and eyespot size and wing shape were not strongly correlated with each other. Overall, larval growth rate seems to be weakly related to eyespot size and wing shape, indicating that seasonal plasticity in *M. leda* is primarily mediated by other mechanisms.

# INTRODUCTION

As an adaption to the seasonality of their environment, many organisms respond plastically to their environment so that they display seasonal forms (seasonal polyphenism, *Tauber, Tauber & Masaki, 1986*). Insects often show developmental plasticity where individuals use environmental cues during early stages to produce an adult phenotype that matches the environment experienced during the adult stage (*Shapiro, 1976*; *West-Eberhard, 2003*). These cues can for example be temperature, day length, or food quality (*Yoon et al., 2023*). Moreover, organisms can use more than one cue, and these cues may interact with each other (*Yoon et al., 2023*). Parts of the physiological cascade that translates environmental cues into induction of the phenotype are well understood in some insect model organisms (*Baudach & Vilcinskas, 2021*; *Monteiro, 2017*; *Oostra et al., 2011*; *Singh et al., 2020*; *Steward et al., 2022*). However, we have limited insight into how multiple cues are integrated, and induce the expression of a suite of traits, especially in non-model organisms.

A prominent example of adaptive phenotypic plasticity in seasonal environments is the seasonal polyphenism exhibited by many tropical satyrine (Nymphalidae: Satyrinae) butterflies, including most Mycalesina (such as *Bicyclus* and *Mycalesis*) and some Melanitini (*Bhardwaj et al., 2020*; *Braby, 1994*; *Brakefield & Larsen, 1984*; *Halali, Brakefield & Brattström, 2024*; *van Bergen & Oostra, 2023*). These butterflies express a wet-season form with large ventral eyespots along the wing margins, and a dry-season form with reduced eyespots and overall a cryptic wing pattern (*Brakefield & Larsen, 1984*; *Brakefield & Reitsma, 1991*). In addition to color pattern, other traits covary in seasonal forms, including wing shape (*Brakefield, 1987*). While most studies focus on the strong effects of temperature (*Kooi & Brakefield, 1999*; *Nokelainen et al., 2018*; *van Bergen et al., 2017*; *Windig, 1992*), other cues can interact with temperature in determining eyespot size and other traits. For example, the quality of the host plant can interact with temperature in determining eyespot size (*Kooi, Brakefield & Rossie, 1996*; *Singh et al., 2020*). How cues are translated into adult phenotype is well understood for the satyrine butterfly *Bicyclus anynana*. For example, we know how temperature cues induce differences in the temporal expression of ecdysone hormone and certain genes, and how this affects wing pattern development (*Oostra et al., 2011, 2014*; *Tian & Monteiro, 2022*). However, it is mostly unknown how multiple environmental cues can drive wing pattern plasticity together, presumably by converging on the same regulatory system.

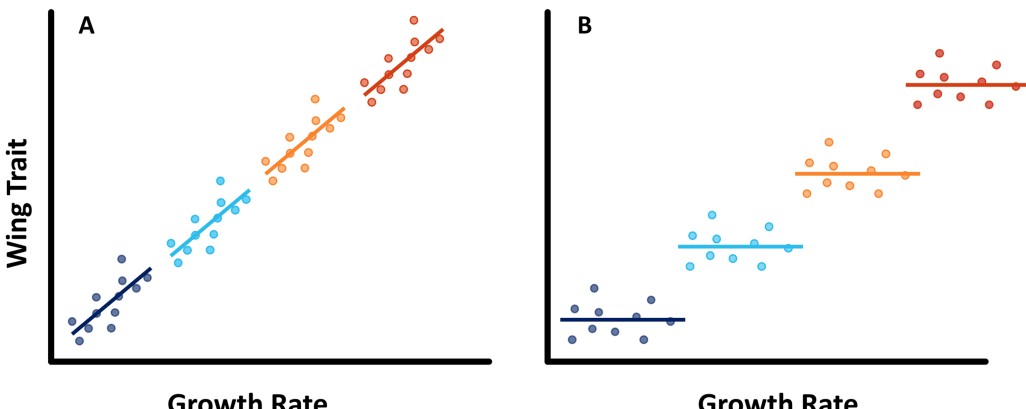

**Figure 1 Conceptual figure.** Predicted regression lines within treatments in case (A) if larval growth rate is part of the pathway that translates the environmental cues into the expression of phenotypes, we would expect to see a relationship between wing traits and growth rate across experimental treatment groups, as well as within treatments, or (B) if larval growth rate is not part of this pathway, we may still see a relationship between larval growth rate and adult phenotype among experimental treatment groups, but no such relationship within treatments.

Larval growth rate could potentially integrate multiple cues for eyespot size in satyrine butterflies. Low temperatures and poor food quality experienced by larvae can induce the development of dry-season form adults (small eyespots) in *Bicyclus* butterflies, and both of these factors also reduce larval growth rate (*Kooi, Brakefield & Rossie, 1996*; *Nokelainen et al., 2016*; *Windig, 1992*, *1994*). Even within a treatment, larval growth may be correlated with adult phenotype (*Windig, 1994*). Therefore, it has been hypothesized that larval growth rate is part of the pathway that translates the environmental cues into the expression of phenotypes (*Kooi, Brakefield & Rossie, 1996*; *Windig, 1992*). However, it is not clear if there is a direct relationship between larval development and adult phenotype. It is also possible that each is correlated with these environmental factors generating a secondary correlation, but without a causal link between them. Moreover, these studies were limited to a small number of closely related species (mainly *B. anynana*) and usually focused on a single metric of adult phenotype: eyespot size.

Here we investigate how multiple cues affect the expression of seasonal polyphenism in the common evening brown butterfly (*Melanitis leda* L., Melanitiini, Satyrinae, Nymphalidae). *M. leda* is a tropical butterfly that is distantly related to *Bicyclus* species (estimated divergence time of 54.36 mya, *Kawahara et al., 2023*), and shows prominent polyphenism in eyespot size as well as wing shape (*Brakefield, 1987*; *Halali et al., 2019*). No experimental studies on its developmental plasticity have been published so far. We tested for the use of temperature and host plant as cues and described provisional reaction norms for two populations. We predicted that *M. leda* would respond to temperature and host plant in the same way as other satyrines: more wet-season phenotypes at higher temperatures and on better host plants. We also performed preliminary experiments to gauge interactions between temperature and humidity, expecting higher humidity to result in more wet-season phenotypes at a given temperature. We then studied the relationship between larval growth rate and two aspects of adult phenotype: eyespot size and wing

shape. In three separate experiments, we manipulated larval growth rate by varying either only rearing temperature, both temperature and relative humidity, or only host-plant species. If growth rate indeed integrates environmental cues and adult phenotype, growth rate and phenotype are predicted to not only be related among treatments, but also within treatments (Fig. 1).

## MATERIALS AND METHODS

### Study organism

*M. leda* is among the most widespread butterflies, distributed from Africa to Asia and Oceania (*Latorre, 2018*). It is able to use a wide variety of grass species (Poaceae) as host plant (*Molleman, Halali & Kodandaramaiah, 2020b*). *M. leda* displays seasonal phenotypic plasticity where wet-season forms have large and conspicuous eyespots, while dry-season forms have small eyespots (Fig. 2). Dry-season forms also have more falcate forewing tips and longer hindwing tails compared to the wet-season forms (Fig. 2; *Brakefield, 1987*; *Halali et al., 2019*). Field data suggest that this seasonal plasticity is to some extent mediated by temperature (*Brakefield, 1987*; *Halali et al., 2021*; *Roy et al., 2021*). Furthermore, the variation among dry-season form wing patterns of *M. leda* represents one of the most multi-faceted polymorphisms in wing coloration found in any animal (*Ruiter & Brakefield, 1994*).

### General procedures and data collection

Three experiments were performed. The first experiment tested the effect of temperature on adult phenotype in a population from Ghana (Temperature Experiment). The second included effects of temperature and humidity using a population from South India (Temperature and Humidity Experiment). The third experiment tested for effect of host-plant species using the South-Indian population (Host Plant Experiment). While some methodological details differ, the data taken from these experiments are comparable.

In most cases, the parental butterflies were held in group cages so that eggs could not be attributed to particular mothers and were divided randomly over the treatments. When we did have eggs from specific mothers, we took special care to divide her offspring evenly over the treatments. In summary, caterpillars were reared on grasses growing in flower pots either from seeds or collected from the field (such as in *Molleman, Halali & Kodandaramaiah, 2020b*). The caterpillars were confined to the plants by placing the plants in fine-mesh sleeves. These sleeves were then kept in environmental test chambers that controlled temperature and humidity and had a 12/12 h light-dark schedule. Plants were watered every other day and were replaced once the majority of leaf material had been consumed or general plant quality had deteriorated. Once larvae reached the 5th instar, sleeves were inspected every 24 h for prepupae and pupae. Prepupae and pupae were kept individually in 100 mL cylindrical transparent containers until eclosion. Prepupae were left attached to the plant whenever possible: using adhesive tape, a piece of the plant where the prepupa was attached was fixed to the side or lid of the container, so that the prepupae could hang freely while pupating. The prepupal stage never exceeded one day, so pupation date of prepupa was the following day. Absorbent paper was placed at the bottom of the

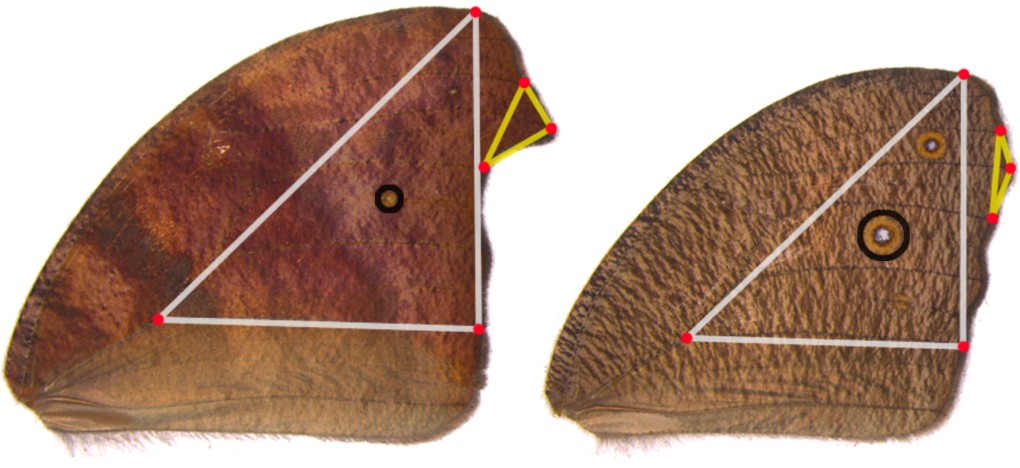

**Figure 2 Landmarks used in this study.** *M. leda* forewings with landmarks used in this study for a dry-season form and a wet-season form from the Ghanaian population. The white triangles denotes the proxy for wing area, the yellow triangles the proxy for wing tip area, and the black circles eyespot area. Photo credit: M. Elizabeth Moore.  

container. Hardened pupae were gently removed from the plant material to be weighed and sexed, and then placed back inside the container. Containers were kept upside down with the absorbent paper on the lid so that pupae and eclosing butterflies were easy to observe through the transparent bottoms. The sides of each of these containers were lined with a strip of paper on which the eclosing adult could climb to enable it to expand its wings. Containers were checked daily for adult eclosions. Butterflies were frozen one day after eclosion.

 Data were collected on development time from egg hatching to pupation (larval development time), and from pupation to eclosion (pupal time). The wings of the enclosed butterflies were removed and photographed or scanned. For the Temperature Experiment, both left and right wings were photographed on the dorsal side, and for the other experiments, one wing in a pair was imaged on the dorsal and the other on the ventral side. Eyespots, forewing tip, and hindwing tails were measured using an ImageJ macro (*Schindelin et al., 2015*) and averaged between left and right wings when appropriate. For this study, we focused on the largest eyespot on the forewing to represent eyespot size and the forewing tip to represent wing shape (Fig. 2). We used particular vein intersections as landmarks for our proxies for wing area and wingtip area (Fig. 2).

## Temperature experiment, Ghanaian population

We collected eggs from 14 females from a stock population established based on 80 females from Bobiri, Ghana (Cites collection and export permit 012068 issued by the Wildlife Division, Ghana), and used at most the third generation reared in the laboratory. Sleeves contained 5–30 larvae, with the average being ca. 20, which were fed ca. two-week-old corn seedlings. These sleeves were placed within environmental test chambers at 75% relative humidity. Larvae were reared at five temperatures: 19 °C, 22 °C, 25 °C, 28 °C and 31 °C. Because of unexpected results, a follow-up experiment was carried out at three

temperatures: 22 °C, 25 °C, and 28 °C (results in Appendix 1). Photographs were taken using a microscope imaging system (Leica DC200 digital camera with Leica MZ12 microscope).

## Temperature and humidity experiment, Indian population

About fifty female *M. leda* butterflies were collected in Thiruvananthapuram, Kerala, South India, to provide eggs, and experiments were performed on three subsequent generations. Sleeves were set up with 15–17 caterpillars each, and were fed ca. two-week-old corn seedlings. Two environmental test chambers were used during subsequent experiments with combinations of four temperatures (19 °C, 21 °C, 27 °C, and 31 °C) and two levels of humidity (65% and 85% relative humidity). However, for technical reasons, a complete set of temperature and humidity combinations was not achieved so that the effect of humidity during rearing on adult phenotype could not be tested formally. The wings of the enclosed butterflies were scanned for phenotypic measurements (Konica Minolta, Bizhub 363, Osaka, Japan).

## Host plant experiments, Indian population

The data presented here are based on the rearing of the fourth batch of larvae described in *Molleman, Halali & Kodandaramaiah (2020b)*. In short, about fifty female *M. leda* butterflies were collected in Thiruvananthapuram to provide eggs. Larvae were then grown on potted plants belonging to 18 species of grass in an environmental test chamber with a constant temperature of 24 °C and a constant humidity of 69%. Photos of the wings of adults were taken with a Nikon D7000 camera under standard light inside a closed white styrofoam box with a shutter speed of 1/125 and an aperture of F14.

## Data analysis

Relative eyespot size was measured by dividing the area of the M3 eyespot by a proxy of total wing area (Fig. 2). Wing shape was quantified as the area of the wing-tip triangle divided by the proxy for wing area (Fig. 2). Larval growth rate was calculated as (pupal mass)$^{1/3}$/larval development time), following *Tammaru & Esperk (2007)*. We first explored the distribution of growth rate, relative eyespot size, and wing shape for both sexes using density plots. While larval growth rate approached a normal distribution, relative eyespot size and wing shape were biased towards very small numbers and this could be corrected by using the square-root transformation. For ease of use, we use 'eyespot size' as a shorthand for 'square root of relative eyespot size' and 'wing shape' as a shorthand for 'square root of relative size of the forewing tip triangle'.

To generate provisional reaction norms, the growth rate, eyespot size and wing shape were plotted against treatment for all three experiments. For the Temperature and Humidity Experiment, data from a pilot experiment were included in the provisional reaction norm. These were second instar larva that had beenreared at 85% RH at either 21 °C or 27 °C, and sex was not determined. For the Host Plant Experiment, plants on which less than eight larvae were reared to adults were excluded, and plants were ordered according to larval growth rate.

To test if growth rate predicts wing traits within treatments, we implemented mixed models and performed model selection based on the Akaike Information Criterion corrected for sample size (AICc). In these models, the dependent variables were either the square root of relative eyespot size or of relative forewing tip size (wing shape). Predictors were larval growth rate, treatment (temperature, humidity, or host plant species), sex, wing area, and interactions between sex and growth rate and sex and treatment when the sample sized allowed it. Sleeve was included as random effect. We made sure that interactions between categorical predictors and growth rate were calculated correctly by subtracting the average growth rate from the measured growth rates in each experiment. To obtain more insight into relationships with body size and development time, we performed further analyses presented in Appendix 3. These include models that did not include wing area as a predictor, and models that replaced the predictor growth rate by pupal mass and development time and their interaction. All analyses were performed using the R packages lme4 (version 1.1.35.3; *Bates et al., 2011*) with lmerTest (version 3.1.3; *Kuznetsova, Brockhoff & Christensen, 2015*) to estimate degrees of freedom, and MuMIn (version 1.47.5; *Barton & Barton, 2015*) to perform model selection in R (version 4.4.0; *R_Core_Team, 2023*), and graphs were made using the R package ggplot2 (3.5.1; *Wickham, 2016*).

# RESULTS

## Cue use and reaction norms

We reared 364 individuals in 44 sleeves in the Temperature Experiment, 82 individuals in 18 sleeves in the Temperature and Humidity Experiment, and 260 individuals in 93 sleeves in the Host Plant Experiment (see sleeves numbers and sample sizes per treatment and sex in Appendix 2). Therefore, we report on a total of 706 butterflies that were reared from egg to adult.

Larval growth rate generally increased with temperature (Figs. 3A & 3B), although it was similar for those reared at 22 °C and 25 °C in the Temperature Experiment (Fig. 3A). The range of growth rates on different host-plant species was only slightly smaller than that generated by using different temperatures (Fig. 3C). Differences in growth rate between temperature treatments were mainly due to differences in development time, as at low temperature (19 °C) the development time was almost three times longer compared to that at high temperature (31 °C), while pupal mass was about one third higher when caterpillars are reared at lower temperatures (Appendix 3). In contrast, within treatments, most of the variation in growth rate was caused by variation in pupal mass, while development-time variation within treatments was modest compared to among treatment variation, especially at higher temperatures (Appendix 3). Across treatments, higher pupal mass was associated with longer development time, but within treatments, higher pupal mass was generally associated with shorter development (*i.e.*, reaction norms for age and size at maturity had a negative slope). Pupal mass was strongly correlated with wing area, but this relationship differed between temperature treatments and the sexes (Appendix 3). Butterflies tended to be smaller when they were reared at higher temperatures (with high growth rate), while they tended to be larger on plants on which they grew faster.

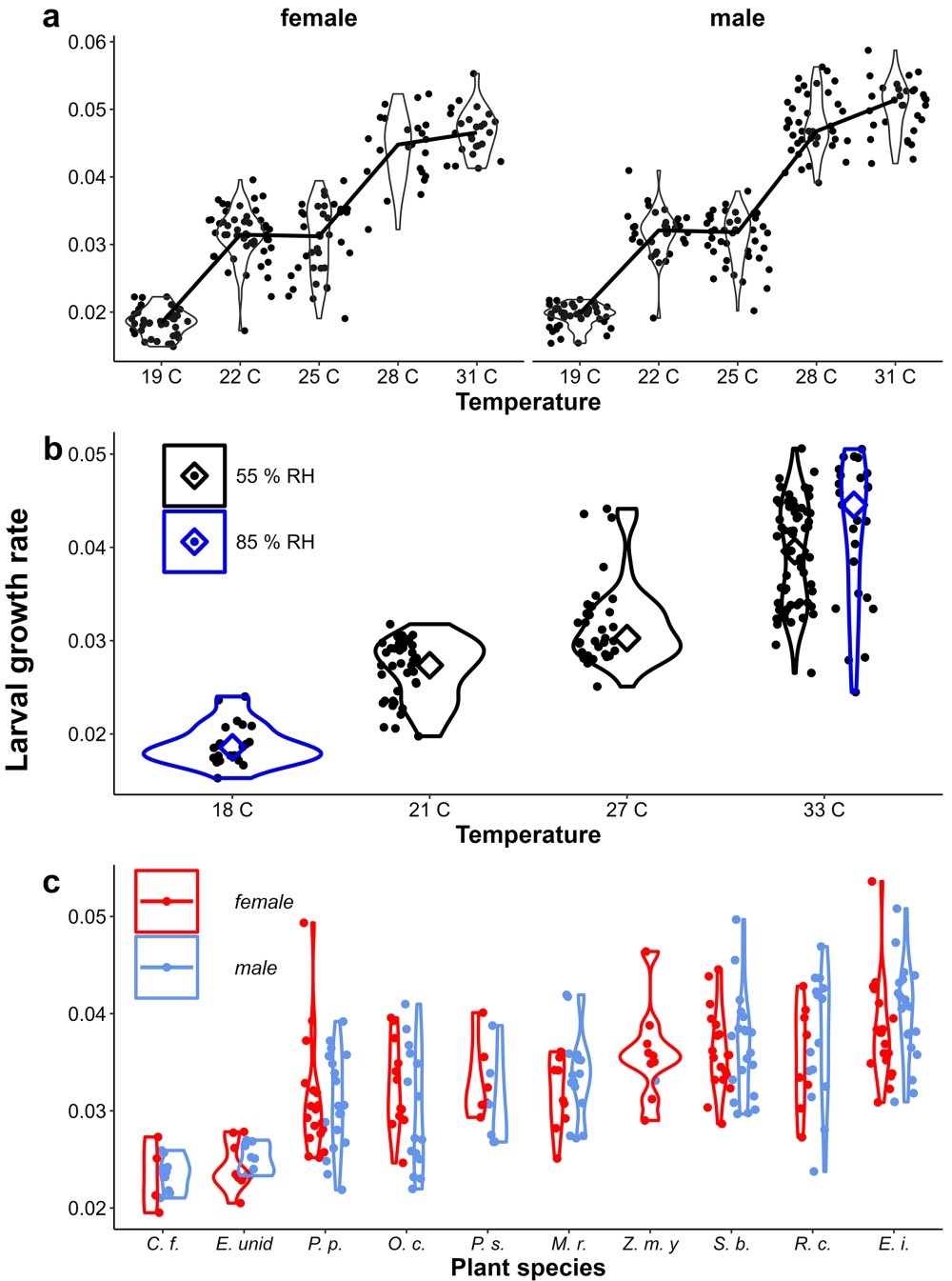

**Figure 3 Provisional reaction norms of *M. leda* growth rate.** Provisional reaction norms of *M. leda* growth rate for (A) the Temperature Experiment, (B) the Temperature and Humidity Experiment, and (C) the Host-Plant Experiment. Larval growth rate was calculated as (pupal mass)$^{1/3}$ / (larval development time). Violin plots represent mirrored density functions illustrating the distribution of eyespot size within each treatment. For (A) the average of eyespot size within each treatment is connected with a line. For (C) only plants with N > 7 were included and plants were sorted from low to high larval growth rate. No true dry-season form butterflies were produced in the Host-Plant Experiment so the y-axis was truncated. Plant species are: *C. f.* = *Cymbopogon flexuosus* (NeesexSt.) Watson, *E. unid* = Unidentified like *Eleusine*, *P. p.* = *Pennisetum polystachion* (L.) Schult., *O. c.* = *Oplismenus compositus* (L.), *P. s.*= *Paspalum scrobiculatum* L., *M. r.* = *Melinis repens* (Willd.) Zizka, *Z. m. y* = *Zea mays* L. seedlings, *S. b.* = *Setaria barbata* (Lam.) Kunth, *R. c.* = *Rottboellia cochinchinensis* (Lour.) Clayton, *E. i.* = *Eleusine indica* (L.) Gaertn.                                     

Eyespot size had a bimodal distribution across all data, but was unimodal within treatments (Appendix 2, Figs. 4A & 4B). Eyespot size increased with increasing rearing temperature in both populations (Figs. 4A & 4B). However, the relationship was highly non-linear in the Ghanaian population with eyespot size being smaller when reared at 25 °C than at 27 °C (Fig. 3A). Similar results were found in the follow-up experiment designed to verify this non-linear pattern (Appendix 1). Notably, the sexes showed qualitatively similar temperature reaction norms for eyespot size (Figs. 4A, 4B) that were nevertheless significantly different in the Temperature Experiment (Table 1, interaction between temperature and sex). At intermediate rearing temperatures, individuals with both wet- and dry-season form eyespots were produced, with a few intermediates (22 °C, 25 °C Fig. 3A, 21 °C, 27 °C Fig. 4B). Higher humidity appeared to result in more wet-season form individuals at intermediate temperatures (Fig. 4B), but note that these are pilot data starting from 2nd and 3rd instar caterpillars, and these data are not included in further analyses. Average eyespot size also varied with host-plant species (Fig. 4C), but the range was narrower as full dry-season phenotypes were not produced in this batch (other batches without quantitative data on wing phenotypes did include dry-season phenotypes). Larger wings were associated with relatively smaller eyespots, indicating that wet-season form butterflies tend to be smaller than dry-season form ones (see Appendix 3 for an illustration of this relationship and graphical and statistical analyses with development time and pupal mass as predictors of butterfly phenotype).

In contrast to eyespot size, wing shape did not show a bimodal distribution in the Indian population, as a large number of individuals showed intermediate wing shapes (Fig. 5, Appendix 2). Otherwise, results for wing shape were similar to those for eyespot size: increasing temperature caused more wet-season phenotypes (less falcate wing tips), and there was an effect of host plant (Fig. 5), with a significant interaction between temperature and sex in the Temperature Experiment (Table 1). Larger wings were also associated with more falcate wing tips (see Appendix 3 for an illustration of this relationship and graphical and statistical analyses with development time and pupal mass as predictors of butterfly phenotype). While eyespot size and wing shape were correlated with each other, this relationship was not particularly strong (Fig. 6), with some butterflies combining large eyespots with falcate wing tips and *vice versa* (Fig. 7).

## Growth rate influencing eyespot size

At higher temperatures, larval growth was faster and adults tended to show more wet-season phenotypes (larger eyespots) in both the Ghanaian population (Fig. 8A, Table 1) and in the Indian population (Fig. 8B, Table 1). Similarly, higher growth rates were associated with larger eyespots within treatments (Fig. 8C, Table 1). However, within treatments, the relationship between growth rate and eyespot size was not very strong, with much scatter and in some treatments opposite trends (Fig. 8, see for statistics of individual regression lines Appendix 2). The relationship also differed between the sexes (Table 1), and in the Host Plant Experiment this relationship was much stronger among males (Fig. 8C). Moreover, at intermediate temperatures, there were clear differences in adult phenotype between temperature treatments that produced a similar range of growth rates

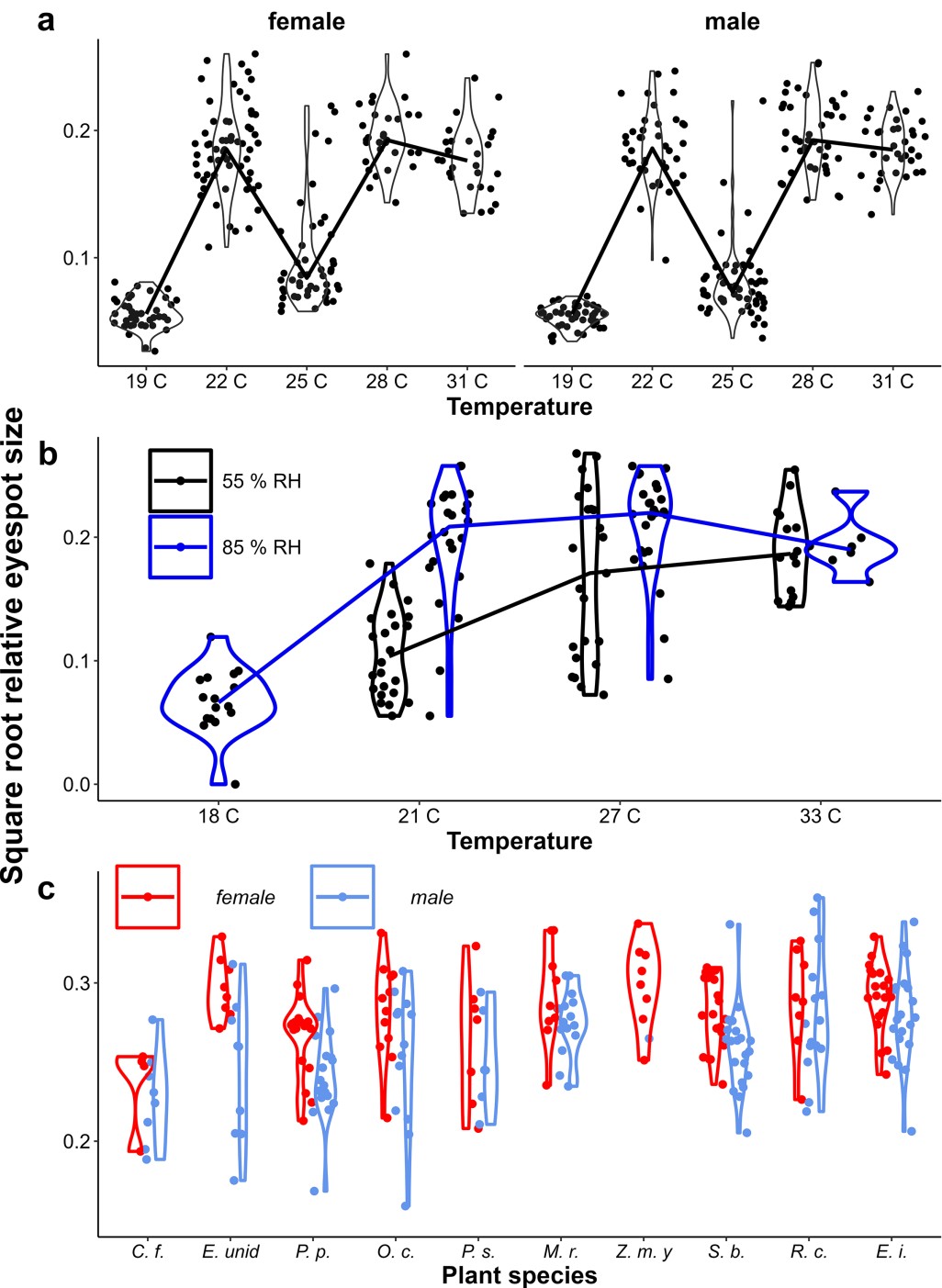

**Figure 4 Provisional reaction norms of *M. leda* eyespot size.** Provisional reaction norms of *M. leda* eyespot size for (A) the Temperature Experiment, (B) the Temperature and Humidity Experiment, and (C) the Host-Plant Experiment. Eyespot size was calculated as the square root of the area of the eyespot divided by the proxy for wing area (Fig. 2). Violin plots represent mirrored density functions illustrating the distribution of eyespot size within each treatment. For (A and B) the average of eyespot size within each treatment is connected with a line. Note that for (B) the data for high humidity for temperatures 21 °C and 27 °C are from a pilot experiment starting from 2nd instar caterpillars and sex was not recorded. For (C) only plants with N > 7 were included and plants were sorted from low to high growth rate. No true dry-season form butterflies were produced in the Host-Plant Experiment so the y-axis was truncated. Full species names are given with Fig. 3.           

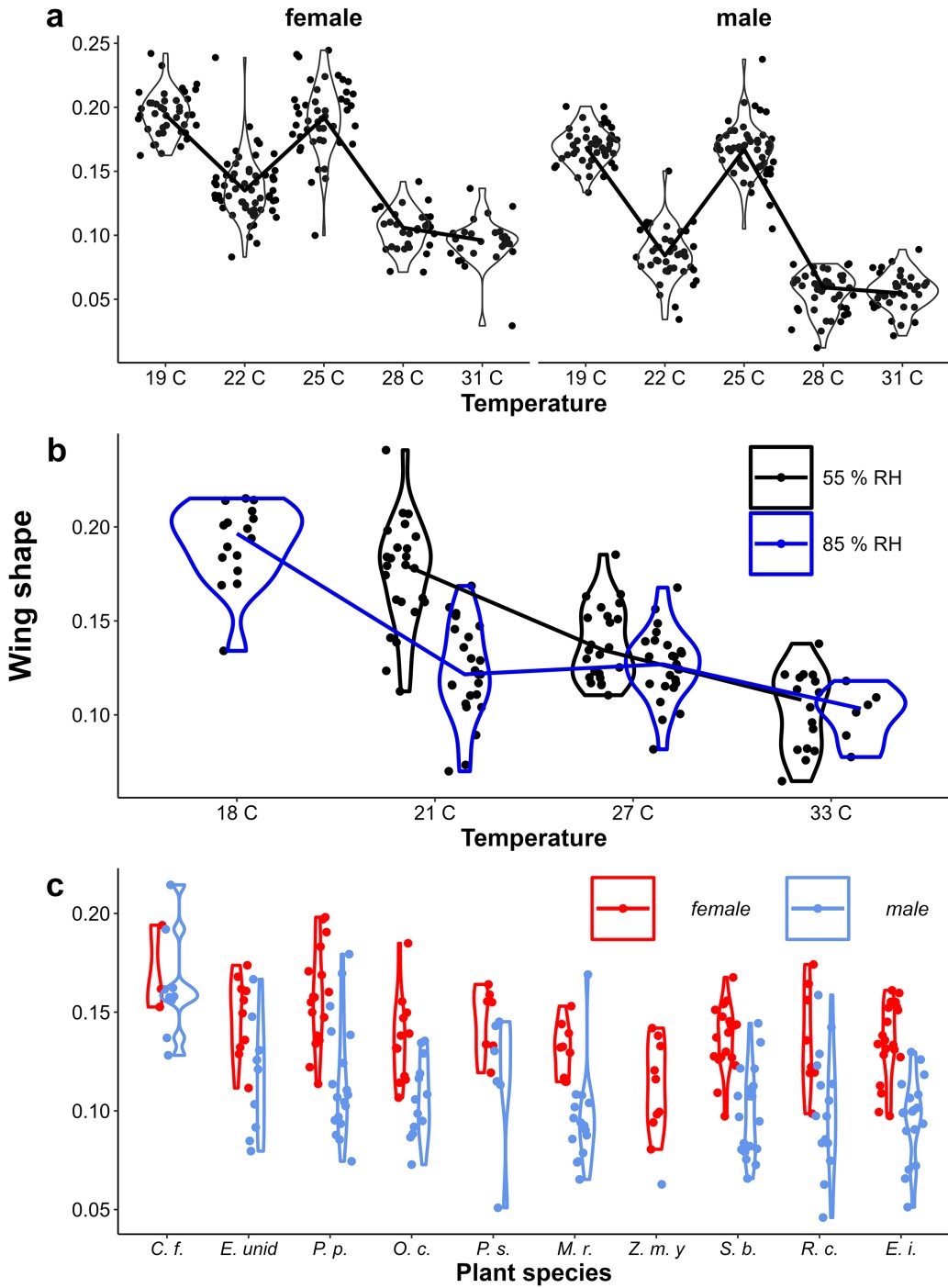

**Figure 5 The effect of larval growth rate on wing shape.** Provisional reaction norms of *M. leda* wing shape for (A) the Temperature Experiment, (B) the Temperature and Humidity Experiment, and (C) the Host-Plant Experiment. Wing shape was calculated as the square root of the area of the wing tip area divided by the proxy for wing area (Fig. 2). Note that for (B) the data for high humidity for temperatures 21 °C and 27 °C are from a pilot experiment starting from 2nd instar caterpillars and sex was not recorded. For (C) only plants with N > 7 were included and plants were sorted from small to large average eyespot size. Full species names are given with Fig. 3.    

**Table 1 Results of linear regression for wing shape.** Results of mixed models (Type III Analysis of Variance Table with Satterthwaite's method) of effects of growth rate and other predictors on the square root of relative eyespot size and the wing shape index with a) full models, and b) models selected based on AICc values. For eyespots in the Temperature experiment, the sleeve random effect explained no variation and was excluded from the model (linear regression). Temp = temperature treatment (categorical), SoS = sum of squares, MS = mean squares, NumDF = degrees of freedom of numerator, DenDF = degrees of freedom of denumerator, Models = number of models in the top five that included the predictor. Bold font indicates $p < 0.05$.

**a) Full models**

**Square root relative eyespot size** · **Wing shape**

**Temperature experiment, N = 364, sleeves = 44**

| | SoS | MS | NumDF | DenDF | F-value | P-value | SoS | MS | NumDF | DenDF | F-value | P-value |
|---|---|---|---|---|---|---|---|---|---|---|---|---|
| Growth rate | 0.009 | 0.009 | 1 | 351.0 | 13.12 | **<0.001** | 0.014 | 0.014 | 1 | 307.2 | 54.7 | **<0.001** |
| Temperature | 0.294 | 0.073 | 4 | 351.0 | 110.3 | **<0.001** | 0.047 | 0.012 | 4 | 40.8 | 46.4 | **<0.001** |
| Sex | 0.005 | 0.005 | 1 | 351.0 | 8 | **0.004** | 0.000 | 0.000 | 1 | 350.1 | 0.54 | 0.462 |
| Wing area | 0.019 | 0.019 | 1 | 351.0 | 28 | **<0.001** | 0.027 | 0.027 | 1 | 350.9 | 109 | **<0.001** |
| Temp: sex | 0.015 | 0.004 | 4 | 351.0 | 5.50 | **<0.001** | 0.001 | 0.000 | 4 | 338.9 | 1.15 | 0.334 |
| Growth rate: sex | 0.007 | 0.007 | 1 | 351.0 | 10.94 | **0.001** | 0.000 | 0.000 | 1 | 349.7 | 0.38 | 0.537 |

**Temperature and humidity experiment, N = 82, sleeves = 18**

| | SoS | MS | NumDF | DenDF | F-value | P-value | SoS | MS | NumDF | DenDF | F-value | P-value |
|---|---|---|---|---|---|---|---|---|---|---|---|---|
| Growth rate | 0.006 | 0.006 | 1 | 76.0 | 4.62 | **0.035** | 0.003 | 0.003 | 1 | 76.6 | 6.43 | **0.013** |
| Temperature | 0.008 | 0.003 | 3 | 20.2 | 2.28 | 0.110 | 0.001 | 0.000 | 3 | 15.2 | 0.43 | 0.737 |
| Wing area | 0.021 | 0.021 | 1 | 76.7 | 16.9 | **<0.001** | 0.02024 | 0.0202 | 1 | 74.5 | 50.6 | **<0.001** |

**Host plant experiment, N = 260, sleeves = 93**

| | SoS | MS | NumDF | DenDF | F-value | P-value | SoS | MS | NumDF | DenDF | F-value | P-value |
|---|---|---|---|---|---|---|---|---|---|---|---|---|
| Growth rate | 0.006 | 0.006 | 1 | 222.2 | 8.96 | **0.003** | 0.010 | 0.010 | 1 | 201.1 | 31.2 | **<0.001** |
| Plant | 0.015 | 0.001 | 16 | 79.5 | 1.55 | 0.105 | 0.012 | 0.001 | 16 | 66.4 | 2.38 | **0.007** |
| Sex | 0.008 | 0.008 | 1 | 230.0 | 13.4 | **<0.001** | 0.013 | 0.013 | 1 | 226.6 | 39.34 | **<0.001** |
| Wing area | 0.000 | 0.000 | 1 | 236.6 | 0.52 | 0.470 | 0.012 | 0.012 | 1 | 229.8 | 36.0 | **<0.001** |
| Growth rate: sex | 0.003 | 0.003 | 1 | 233.0 | 5.10 | **0.025** | 0.001 | 0.001 | 1 | 233.1 | 3.89 | **0.050** |

**b) Selected models**

**Square root relative eyespot size** · **Wing shape**

**Temperature experiment, N = 364, sleeves = 44 excluded**

| Delta = 6.0 | Models | SoS | MS | NumDF | DenDF | F-value | P-value | Models | SoS | MS | NumDF | DenDF | F-value | P-value |
|---|---|---|---|---|---|---|---|---|---|---|---|---|---|---|
| Growth rate | 5 | 0.002 | 0.002 | 1 | 358 | 3.330 | 0.068 | 4 | 0.016 | 0.016 | 1 | 324.9 | 62.26 | **<0.001** |
| Temperature | 4 | 0.526 | 0.131 | 4 | 358 | 174.264 | **<0.001** | 5 | 0.045 | 0.011 | 4 | 41.0 | 44.4 | **<0.001** |
| Sex | 2 | | | | | | | 5 | 0.030 | 0.030 | 1 | 348.7 | 121 | **<0.001** |
| Wing area | 3 | | | | | | | 5 | 0.035 | 0.035 | 1 | 354.7 | 140 | **<0.001** |
| Temp: sex | 1 | | | | | | | 2 | | | | | | |
| Growth rate: sex | 0 | | | | | | | 1 | | | | | | |

**Temperature and humidity experiment, N = 82, sleeves = 18**

| | Models | SoS | MS | NumDF | DenDF | F-value | P-value | Models | SoS | MS | NumDF | DenDF | F-value | P-value |
|---|---|---|---|---|---|---|---|---|---|---|---|---|---|---|
| Growth rate | 5 | 0.018 | 0.018 | 1 | 49.2 | 14.7 | **<0.001** | 4 | 0.009 | 0.009 | 1 | 44.9 | 24.5 | **<0.001** |
| Temperature | 2 | | | | | | | 0 | | | | | | |
| Humidity | 3 | | | | | | | 1 | | | | | | |
| Sex | 0 | | | | | | | 2 | | | | | | |
| Wing area | 5 | 0.025 | 0.025 | 1 | 79.8 | 20.66 | **<0.001** | 5 | 0.023 | 0.023 | 1 | 79.3 | 60.3 | **<0.001** |

**b) Selected models**

**Square root relative eyespot size**        **Wing shape**

**Temperature experiment, $N = 364$, sleeves = 44 excluded**

| Delta = 6.0 | Models | SoS | MS | NumDF | DenDF | F-value | *P*-value | Models | SoS | MS | NumDF | DenDF | F-value | *P*-value |
|---|---|---|---|---|---|---|---|---|---|---|---|---|---|---|
| Temp: sex | 0 | | | | | | | 1 | | | | | | |
| Growth rate: sex | 0 | | | | | | | 0 | | | | | | |
| Host plant experiment, $N = 260$, sleeves = 93 | | | | | | | | | | | | | | |
| Growth rate | 5 | 0.023 | 0.023 | 1 | 162.2 | 36.63 | **<0.001** | 5 | 0.025 | 0.025 | 1 | 169 | 79.7 | **<0.001** |
| Plant | 0 | | | | | | | 0 | | | | | | |
| Sex | 4 | 0.026 | 0.026 | 1 | 238.6 | 41.18 | **<0.001** | 4 | 0.013 | 0.013 | 1 | 241 | 40.8 | **<0.001** |
| Wing area | 2 | | | | | | | 3 | 0.012 | 0.012 | 1 | 247 | 38.2 | **<0.001** |
| Growth rate:sex | 2 | 0.002 | 0.002 | 1 | 247.9 | 4.53 | **0.034** | 2 | 0.001 | 0.001 | 1 | 244 | 4.32 | **0.039** |

(22 °C and 25 °C in Fig. 3A, and 21 °C and 27 °C in Fig. 8B). In the host plant experiment however, when growth rate was included in the model, host plant was not selected in the model and was not a significant predictor, indicating that the effect of host plant on eyespot size can be explained by its effect on growth rate. Wing area was a significant predictor of relative eyespot size in all three experiments (Table 1A) and was selected into the final models (Table 1B). When wing area was included into the models, growth rate remained a significant predictor of relative eyespot size (Table 1; model results without wing area are given in Appendix 3, Table A3.2).

## Growth rate influencing wing shape

In all experiments, lower larval growth rate was associated with more falcate wings in both sexes across and within treatments (Fig. 9, Table 1). However, at intermediate temperatures, there were clear differences in adult phenotype between temperature treatments that produced a similar range of growth rates (22 °C and 25 °C in Fig. 4A, and 21 °C and 27 °C in Fig. 9B), and there was considerable scatter and some treatments showed opposite trends (see Appendix 1 for statistics of individual regression lines). The relationship was stronger among males than among females, particularly in the Host Plant Experiment (Table 1, Fig. 9C). Males tended to have less falcate wings when they were wet-season forms than females, so that the range of wing shapes was wider in males (most clearly seen in Figs. 9A & 9C). Overall, both eyespot size and wing shape were significantly but weakly related to larval growth rate within treatments (Table 1), the relationships appearing stronger for wing shape and among males (Table 1, Appendix 2). Notably, growth rate was a particularly poor predictor of wing phenotype among the 22 °C and 25 °C treatments of the Temperature Experiment. Nevertheless, in the host plant experiment, host plant was a significant predictor, but was not selected in the model when growth rate was included, indicating that the effect of host plant on wing shape size can to a large extent be explained by its effect on growth rate. Wing area was a significant

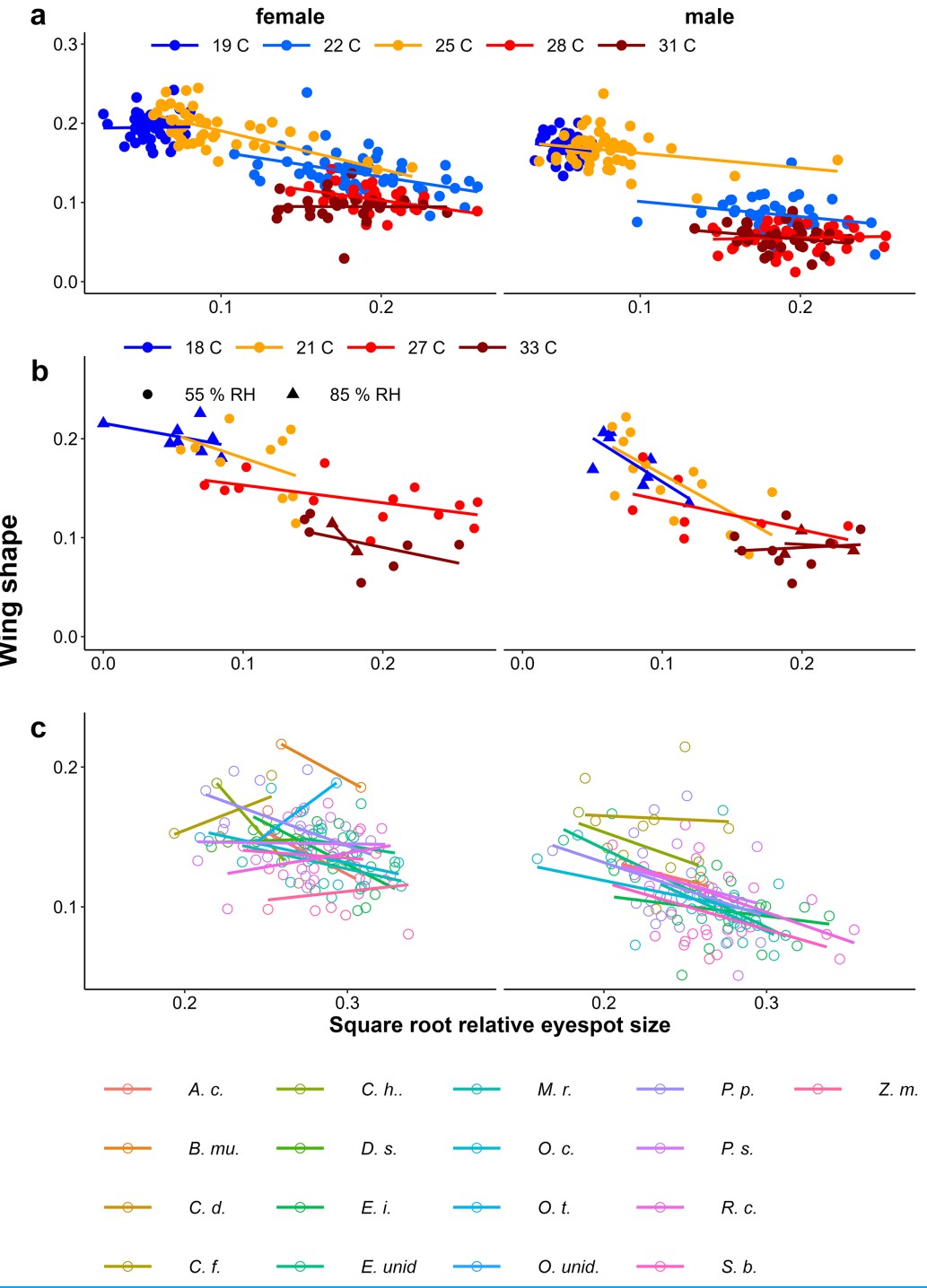

**Figure 6 Scatter plots of eyespot size and wing shape.** Relationship between relative eyespot size and wing shape proxies for (A) the Temperature Experiment, (B) the Temperature and Humidity Experiment, and (C) the Host-Plant Experiment. For (C) only plants with N > 7 were included and plants were sorted from small to large average eyespot size. Full species names are given with Fig. 3.

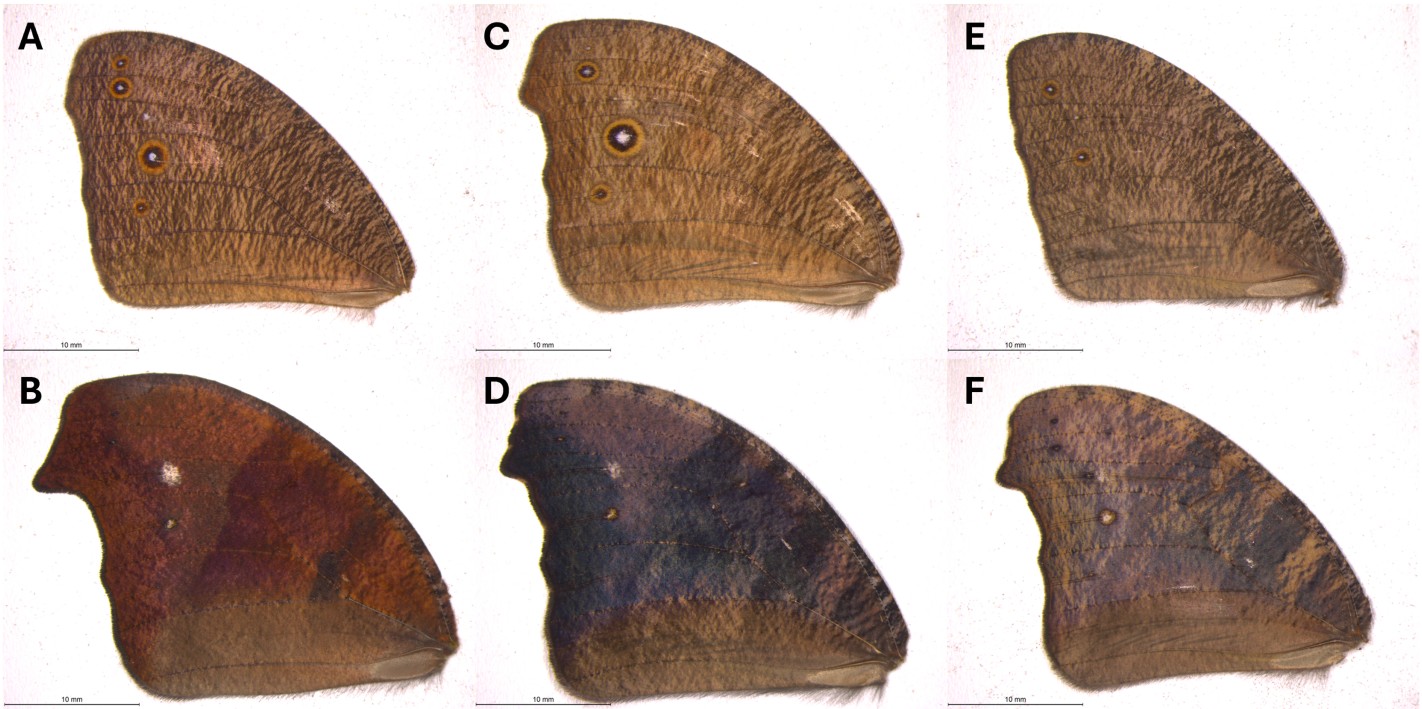

**Figure 7 (A–F) Examples of combinations of eyespot size and wing shape.** *M. leda* wing pattern and shape vary between and among phenotypes. In the more extreme phenotypes (A, B), wet season wings have smooth wing margins, large eyespots, and irrorated background patterns, while dry season wings have falcate wing tips, small eyes, and cryptic coloration. Wing shape and eyespot size are not strongly correlated with each other, resulting in wet-season colorations with prominent falcate tips (C) and/or small eyespots (E), and dry season colorations with smoother margins (D, F), larger eyespots (F), or interspersed irrorated coloration (D, F). Scale bars = 10 mm. Photo credit: M. Elizabeth Moore.

predictor of relative eyespot size in the Temperature Experiment and the Temperature and Humidity Experiment (Table 1A) but was not selected into final models (Table 1B). When wing area was included, growth rate remained a significant predictor of wing shape (model results without wing area are given in Appendix 3, Table A3.2).

## DISCUSSION

We reared more than 700 larvae of *M. leda* under various temperature, humidity, and host-plant conditions. Overall, larvae reared at higher temperatures tended to grow faster and produce more wet-season form phenotypes (larger eyespots and less falcate wing tips), and host-plant species also affected adult phenotype, similar to what has been found in some *Bicyclus* butterflies. *M. leda* attained larger body sizes when reared at lower temperatures and when reared on plants on which they had higher growth rates. Within treatments, the reaction norms for age and size at maturity had a negative slope, with longer development times being associated with smaller body size. Eyespot size and wing shape were not tightly correlated with each other. Even though larval growth rate was consistently a significant predictor of wing phenotype, the relationship did not appear to be strong: there were large differences in adult phenotype between treatments that produced a similar range of growth rates. However, the effect of host plant on wing

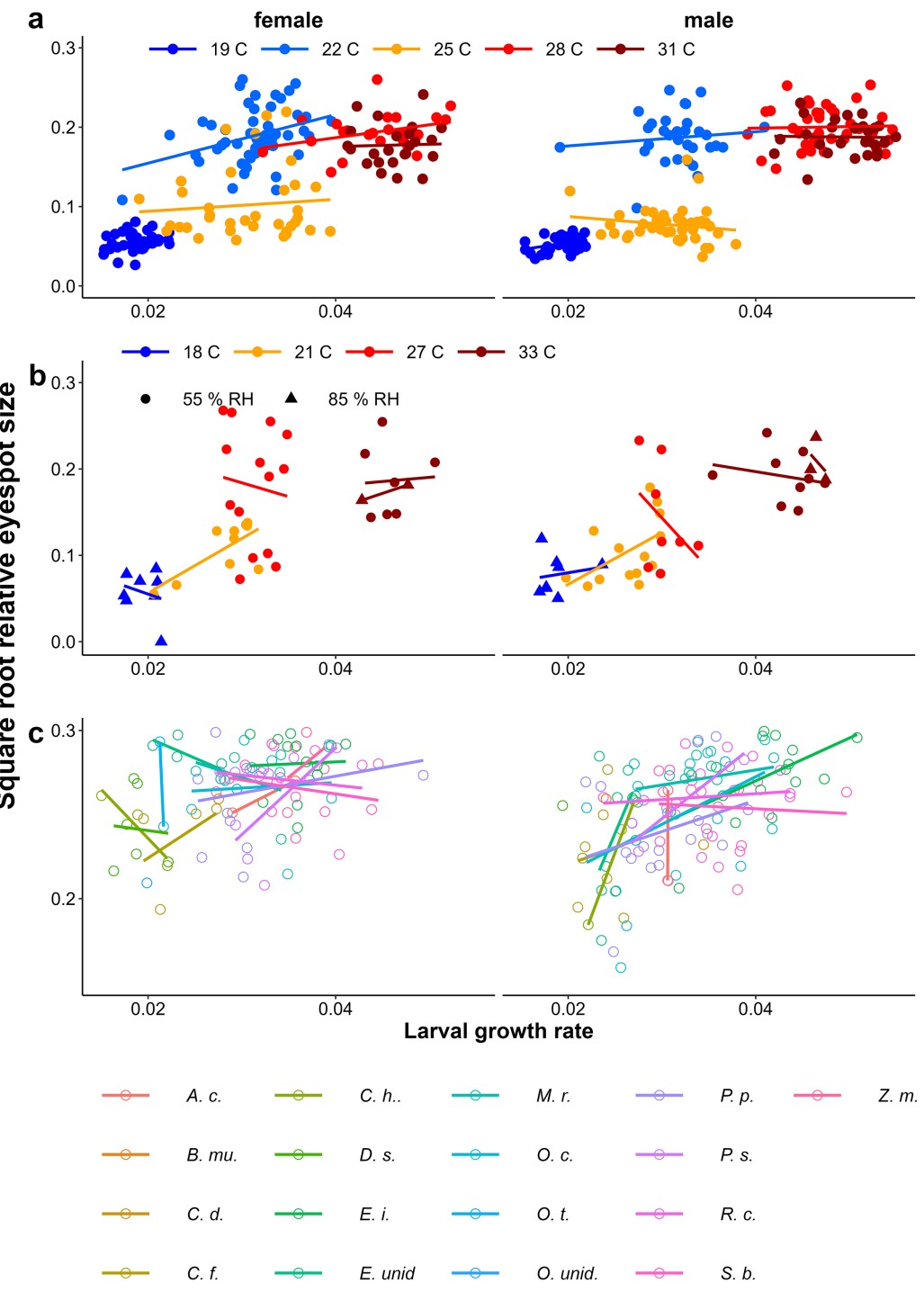

**Figure 8 Relationship between growth rate and eyespot size.** The relationship between growth rate and eyespot size with (A) the Temperature Experiment with the Ghanaian population, (B) the Temperature and Humidity Experiment with the Indian population, and (C) the Host-Plant Experiment with the Indian population. Lines depict linear regressions within treatments as in Fig. 1. See Table 1 for sample sizes, statistical results, and full species names are given with Fig. 3.

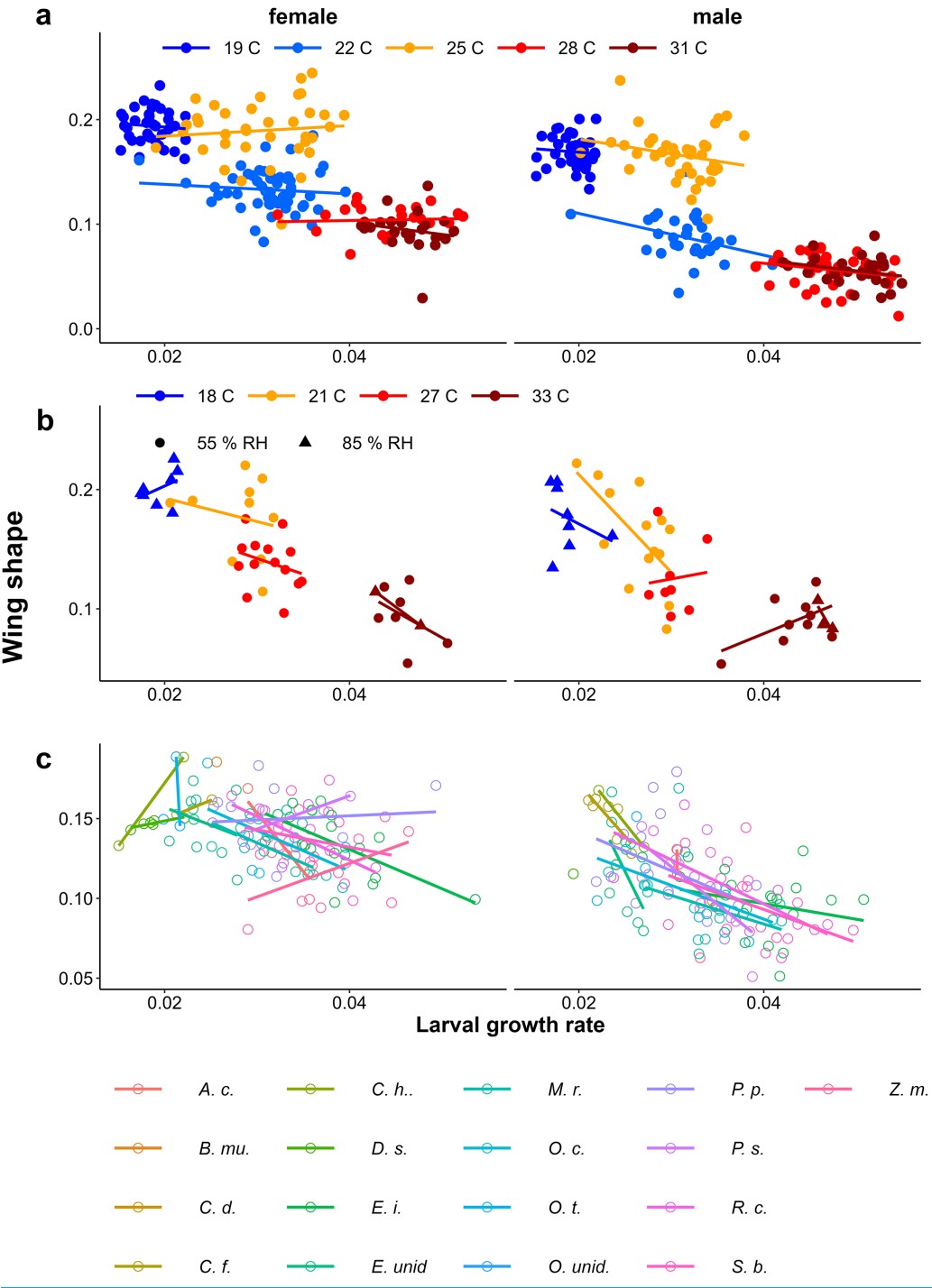

**Figure 9 Relationship between larval growth rate and wing shape.** Scatter plots of larval growth rate and wing shape for (A) the Temperature experiment (Ghana population), (B) the Temperature and Humidity Experiment (Indian population), and (C) the Host-Plant Experiment (Indian Population). Lines depict linear regressions within treatments as in Fig. 1. See Table 1 for sample sizes, statistical results, and full species names are given with Fig. 3.

phenotype appeared to be explained well by larval growth rate. We also found differences between the sexes in the relationship between growth rate and wing phenotype.

The effects of treatments on development time and body size and thus growth rate followed the usual pattern of faster development at higher temperatures and the temperature body size rule (*Atkinson, 1994*) also known as 'hotter is smaller' (*Kingsolver & Huey, 2008*). This may be linked to the seasonal polyphenism as dry-season forms are thought to be larger and have greater fat reserves to be better able to survive dearth periods (*Pijpe, Brakefield & Zwaan, 2007*), while wet-season forms may be smaller to reduce generation time (*Kingsolver & Huey, 2008*). Indeed, larger wings tended to be more falcate, and wing area was often a significant predictor of eyespot size and wing shape. On plants on which larvae grow faster, they tended to attain higher body sizes, and short development time was also associated with larger body size within treatments across all experiments. This resembles the classic reaction norm for age and size at maturity with a negative slope (*Teder, Vellau & Tammaru, 2014*) where some individuals manage to grow quickly and become large, while others are growing slowly and remain small. Variation in growth rate among individuals within a treatment may be attributed to genetic differences between individuals, and effects of environmental differences. In our experiments, the parentage of the eggs was not controlled for, and even full kin would differ genetically. Notably, even under the most favourable conditions, larval survival (from egg-hatching to adult eclosion) hardly exceeds fifty percent in our study species (*e.g.*, on average 8.3 butterflies enclosed from 20 caterpillars per sleeve in the Temperature Experiment, see for more information about survival *Molleman, Halali & Kodandaramaiah, 2020b*). Such large fitness differences between individuals are common, even among clonal organisms (*Schaible, Sussman & Kramer, 2014*). Apart from genetic differences, larval survival and growth rate might depend on non-genetic factors like egg size, egg-quality, the quality of the first bites of food, competitive interactions, and the microbes that colonize them. Future studies may determine the causes of variation in growth rate between individuals.

*M. leda* larvae are more likely to develop into wet-season form adults when reared at higher temperatures, and on host-plant species on which they grow faster. The opposite relationship between body size and wing phenotype between the Temperature Experiment and the Host Plant experiment is not consistent with a simple allometric relationship (Appendix 3). Moreover, the effect of growth rate remains when wing area is included in models for eyespot size an wing shape. That higher temperatures induced wet-season phenotypes similar to distantly related mycalesines (*van Bergen et al., 2017*) may be explained by the increases in temperature as the wet season is approaching and decreases in temperature at the end of the wet season (*Halali et al., 2021*; *van Bergen & Oostra, 2023*). Temperature during larval development is thus likely to be predictive of the season the adult will experience. Using host-plant quality as a cue may be adaptive because towards the end of the wet season, plants will often be of lower quality, and plant quality can thus be a reliable predictive cue for larvae: low plant quality indicating that the dry season is approaching. Conversely, once the rains have started, caterpillars will on average experience high food quality on which they can achieve a faster growth rate. Since butterflies will probably start laying eggs at the beginning of the wet season, plant quality is

probably a reliable cue during this period. Further studies may test whether the developmental plasticity for these two cues and two traits was inherited from a common ancestor (of Mycalesini and Melanitini) or resulted from parallel evolution (*Bhardwaj et al., 2020*).

This climate pattern is common in the tropics but does not apply everywhere. Therefore, the response to temperature might vary between populations (*Roskam & Brakefield, 1996*), similar to findings in *B. anynana* and *B. safitza* (*de Jong et al., 2010*; *Nokelainen et al., 2018*). While the Indian and Ghanaian populations responded qualitatively similarly to temperature in our experiments, a quantitative comparison among the populations would require rearing them together in a common garden experiment. Furthermore, the reaction norms should be considered as provisional. For example, possible effects of environmental test chamber or outdoor weather conditions cannot be excluded in our results. In particular, the dip in the temperature reaction norm for the Ghanaian population, and the effects of humidity need to be investigated further.

The significant effect of growth rate on wing phenotype within treatments may suggest that different cues might in part be integrated through growth rate for wing shape and eyespot size. However, larval growth does not appear to be solely responsible for determining adult phenotype: the relationships within treatment were not particularly strong, and at intermediate temperatures, growth rates were similar while adult phenotypes differed greatly among temperature treatments. Perhaps growth rate during a small section of the development time is the actual cue (a critical period, *Kooi & Brakefield, 1999*). Since larval development may be slow during the first instars on certain plants, but then accelerate in later instars, overall growth rate might be a poor predictor of adult phenotype if there is such a critical period. However, this would not explain the large differences in adult phenotype between treatments that produced nearly identical growth rates such as at 22 °C and 25 °C in the Temperature Experiment. Notably, growth rate was a better predictor of eyespot size and wing shape than host plant species, so that effects of host plant may be mediated by larval growth rate. Overall, growth rate is probably not the main determining factor affecting adult phenotype.

It may be surprising that eyespot size and wing shape appear to be regulated differently (low correlation within individuals). This leads to individuals with a mix of dry and wet-season phenotypic traits such as large eyespots combined with falcate wing tips. In other satyrines, multiple plastic traits tend to be linked, but there are also exceptions (*Mateus et al., 2014*; *van Bergen et al., 2017*). While such disintegration of traits may be expected to be disadvantageous, it could also contribute to the extensive phenotypic variation among individuals (*Ruiter & Brakefield, 1994*) that may hamper search-image formation in its predators (*Cook, 2017*; *Forsman, 2015*; *Forsman, Betzholtz & Franzen, 2015*).

We found notable sex differences in the relationship between larval growth rate and adult phenotype. Perhaps females often reached the limit of maximum eyespot size when they are wet-season form, truncating their size distribution and thus limiting their potential response to growth rate, while in males, eyespots tend to be smaller and to cover a wider range of sizes (see Fig. 3C). For wing shape, the effects of growth rate also tended to

be more pronounced in males than in females. Specifically, males tended to have even less falcate wings when they were wet-season forms than females. This indicates some subtle sexual dimorphism in wing shape in wet-season forms that may be connected to sex-biased movement patterns (*Berwaerts, Van Dyck & Aerts, 2002*) or sexual selection (*Kemp, 2002*; *Molleman, Halali & Kodandaramaiah, 2020a*).

## CONCLUSIONS

We found that cue use of *M. leda* is similar to that of distantly related satyrines such as *B. anynana*. The mechanism of seasonal dimorphism in *M. leda* can be understood as developmental plasticity using temperature and certain aspects of host-plant quality as cues, where larvae reared at higher temperatures and on host plants on which they can grow faster tend to show more wet-season adult phenotypes (larger eyespots and less falcate wing tips). This cue use may in part be mediated by larval growth rate, but seasonal plasticity in *M. leda* appears primarily mediated by other mechanisms that need to be investigated in future.

## ACKNOWLEDGEMENTS

We thank Kwaku Aduse-Poku for help in obtaining the population from Ghana.

### Funding

Funding for the work was provided by the University of Cambridge, the Indian Institute of Science Education and Research Thiruvananthapuram, and the Narodowe Centrum Nauki (National Science Centre, Poland) grant 2021/43/B/NZ8/00966. The funders had no role in study design, data collection and analysis, decision to publish, or preparation of the manuscript.

### Grant Disclosures

The following grant information was disclosed by the authors:
University of Cambridge, the Indian Institute of Science Education and Research Thiruvananthapuram.
Narodowe Centrum Nauki (National Science Centre, Poland): 2021/43/B/NZ8/00966.

### Competing Interests

The authors declare that they have no competing interests.

### Author Contributions

- Freerk Molleman conceived and designed the experiments, performed the experiments, analyzed the data, prepared figures and/or tables, authored or reviewed drafts of the article, and approved the final draft.
- M. Elizabeth Moore conceived and designed the experiments, performed the experiments, analyzed the data, prepared figures and/or tables, authored or reviewed drafts of the article, provided the photos for Figs. 2 and 7, and approved the final draft.

- Sridhar Halali conceived and designed the experiments, performed the experiments, analyzed the data, authored or reviewed drafts of the article, and approved the final draft.
- Ullasa Kodandaramaiah conceived and designed the experiments, authored or reviewed drafts of the article, and approved the final draft.
- Dheeraj Halali performed the experiments, authored or reviewed drafts of the article, and approved the final draft.
- Erik van Bergen conceived and designed the experiments, authored or reviewed drafts of the article, design ImageJ code for measuring wing phenotypetype, and approved the final draft.
- Paul M. Brakefield conceived and designed the experiments, authored or reviewed drafts of the article, and approved the final draft.
- Vicencio Oostra conceived and designed the experiments, analyzed the data, prepared figures and/or tables, authored or reviewed drafts of the article, and approved the final draft.

### Field Study Permissions

The following information was supplied relating to field study approvals (*i.e.*, approving body and any reference numbers):

The Cites collection and export permit for the specimens was issued by the Wildlife Division of Ghana.

### Data Availability

The code and data are available in the Supplemental Files.

### Supplemental Information

Supplemental information for this article can be found online at http://dx.doi.org/10.7717/peerj.18295#supplemental-information.

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
