# Peer review of "Larval growth rate is not a major determinant of adult wing shape and eyespot size in the seasonally polyphenic butterfly *Melanitis leda"

_PeerJ, doi:10.7717/peerj.18295_

## Round 0.1 · original submission · Major Revisions

Thanks for submitting your work to PeerJ. Please change your manuscript as reviewers' comments.

Reviewer 1 ·

Basic reporting

Overall, I found this manuscript quite well written and structured. It represents a complete study testing whether larval growth rate provides an internal cue mediating the effect of different environmental cues on plasticity for a butterfly species. It does an excellent job of placing this research in its broader context.
The figures are mostly clear although more detail is needed in the captions. Particularly, for all figures, please make clear in the caption what the lines represent because they mean very different things in figures 3 and 4 versus figures 5 and 6. For figures 3 and 4, you should also make clear how the “violins” were generated because some readers may not be familiar with violin plots and how to interpret them.
Both the raw data and code have been made available in an appropriate format. I could open all of them and they contained no obvious errors in a basic inspection (although I did not attempt to run them).
That said, some form of readme file should be provided to explain the data file and fully enable its reusability. It should clearly describe the information found in each column, including its units and the conditions under which NA was used (for example, there are cases where pupal time is NA, but there are still wing measurements, so it can’t just be mortality).
Similarly, although the code was reasonably well organized, it was not clear to me what each different file was used for. One had an informative name and a label at the start of the file (wingShapeRegressionPlots), but for the others this was not at all clear. I would recommend giving each file a distinct, informative name and a short description at its start explaining what parts of the analysis/plotting this code was used for (for example, identifying the specific figures produced by number).

Experimental design

This manuscript tests a clearly stated and well-developed hypothesis using three parallel experiments. Although they had some limitations (appropriately acknowledged by the authors), these studies were properly designed to test the authors’ hypotheses and conducted appropriately. The general methods are very clearly described and have most of the details needed for potential replication of the study (see line-by-line comments for a few missing details).
Unfortunately, I am not as happy with the authors’ statistical approach to testing their main hypothesis, and this is where I think the manuscript needs substantial (but feasible) improvement.
Most notably, I am disappointed by Tables 1 and 2 and the authors’ approach to testing a single clear hypothesis by qualitatively comparing a large number of separate statistical tests (in effect, simple vote counting of significance). This results in tables that are difficult to clearly interpret, highly subject to type I or type II error (given the large number of independent tests), and only very weakly account for the large differences in sample sizes between experiments (which affect the statistical power and rate of both forms of error). Interpreting adjusted R2 slightly improves this but still creates quite noisy results that are difficult to interpret. The same approach is applied to the graphical abstract, which is a nice way to visualize Tables 1 and 2 but highlights the difficulty interpreting it and the noisiness of this approach. That the results can be interpreted at all is only because they are likely quite strong, and the authors (appropriately) limit themselves to discussing only the most general trends.
A much more effective and appropriate approach to the statistics would be to find a way to test for the effects of growth rate on a given phenotype using only a single test per experiment. I expect there are multiple approaches that could potentially be used to do this effectively from a properly designed linear/mixed model to structural equation models. A simple approach inspired by your figure 1 would be a linear model with treatment (as a factor to account for the non-linear response), growth rate, sex, and the interaction of growth rate and sex as fixed effects. You don’t need to do any statistical tests on treatment in this model, but by including it and then testing the effects of growth rate its interaction with sex, you effectively test your hypothesis: whether growth rate explains the remaining variation in phenotype after treatment is accounted for. (Another option would be to use the residuals of phenotype after accounting for treatment as the dependent variable AND the residuals of growth rate after accounting for treatment as the independent variable/fixed effect. I believe these two approaches should produce the same results, although incorporating sex could be a little more complicated in this latter case (see https://stats.stackexchange.com/questions/46185/question-on-how-to-normalize-regression-coefficient/46508#46508)). Relatedly, the authors should also change the graphical abstract to reflect this new approach, rather than the previous tables.
It was unclear to me how many sleeves were used per treatment per experiment, but if there were multiple sleeves, this should be a random effect in a mixed model. If it was just one, any sleeve effect is confounded with treatment effect, but this is not a problem for interpreting the residual effect of growth rate specifically.
Somewhat relatedly, the reaction norms for growth rate are never clearly presented graphically or in the text. Some aspects of this can be inferred from figure 5/6, but the specific patterns can be hard to interpret from these figures (especially for the effects of diet). Given the centrality of the reaction norm for growth rate to the author’s hypotheses and the descriptions in various places of how treatments affected growth rate, I believe the paper should have a figure similar to figures 3 and 4, but for larval growth rate, along with a short associated section in the results.
There is also one specific alternative explanation for the wing shape results that could explain why the results for wing shape are stronger than eyespot size: the authors have not addressed the potential for allometric change in wing shape where larger wings have disproportionately larger (or smaller) wing tips. If growth rate is related to overall size (likely, but not guaranteed since it also depends on development time) and size then effects wing shape, this could create some of the relationship between larval growth rate and wing shape. Allometry could be tested directly in multiple ways by relating wing tip shape/area to either mass or overall wing area and checking for the expected relationship. If it is present, it could be accounted for in the model (by adding one of the overall measures of size), or simply acknowledged as a potential cause in the discussion. The same non-linear allometry could also exist for wing spot size and would be worth checking for.
As a side note, the authors do not perform any specific statistical analyses associated figures 3 and 4. This is an unusual approach, but I do not have a problem with this because of how the authors clearly describe these reaction norms as “provisional” and focus on describing only the broadest patterns. In this regard, Appendix 1 is greatly appreciated as further evidence that the unusual reaction norm of the temperature experiment is not due to chance or experimental error, and greatly strengthens this result.

Validity of the findings

The authors interpretation and discussion of their results is appropriate given the limitations noted above of their statistical approach. I think the improvements to their analysis that I request above should, if anything, make these results clearer and better supported, but the validity of the conclusions should be revisited once these changes have been made to confirm they remain valid.

Additional comments

Line by line comments. These are mostly small suggestions for improvements. Those that I think are necessary to address are indicated by *:

Abstract
32: “which are expressed in the dry season” is redundant with “Dry-season forms”, so I think you can drop that phrase from the abstract.
47: It’s not essential, but I think it would help attract readers to have a sentence or so about the implications of this result in the abstract, even if its just something simple like “… indicating that seasonal plasticity in M. leda is primarily mediated by other mechanisms.”

Introduction
49-50: The phrasing of this sentence and combination of “are adapted… using” reads awkwardly. Particularly, while you can seasonal plasticity is an adaptation to seasonal plasticity (in the sense that it has evolved to increase fitness in this context), it is not itself adaptation as a process (since it does not involve heritable change in phenotype). The way this sentence is phrased, it is unclear which claim you are trying to make.
60-62: This is a clear, compelling statement of purpose for the rest of the paper. I really like it.
75: For which cues do we know about this in B. anynana? Just temperature?
88-90: I had a hard time following this sentence. “These environmental factors” (I assume the cues) and “them” (I guess development and phenotype) are a little too ambiguous.
99-100: You may want to be explicit here that these relationships have not been previously tested.

Materials and Methods
117: Since you’ve highlighted earlier the importance of understanding non-model organisms, maybe this sub-section would be better titled “study organism”?
*152-153: Because you moved individuals either as prepupae or pupae (line 141), what time did you count as pupation when determining development time? Was it from this daily check, or did you further check the prepupae to determine the exact day that they pupated? Either is fine, you just in the first case you need to be clear that this is what you defined as pupation.
*153-154: For this photography/scanning, did you first remove the wings from the body, or did you measure the whole butterfly? Did you measure the right wing, left wing, or both?
*159-160: This is a good figure for seeing what you did, but the colors in the caption aren’t right. The circle is white and the area triangle black. Also, either in the figure caption or the main text you should explicitly state what points you used on the wings for the triangles (in this case, it looks like you used specific vein intersections, which you should name if that was the case).
*163: How long had this stock population been in the lab from the field (or was it maintained by repeated addition of new wild females)?
*164: How many sleeves per treatment? This question should be answered for all three experiments.
*165, 173, 184: You use different terms across experiments: environmental test chamber versus incubator. Please use one term consistently or explain the difference. Ideally, you should report the model(s) of chamber used.
*169: As you do for the cameras and scanners, please note the model of imaging system used.
*171: How many females were collected?
181: Just to double check, this data is from the same individuals already reported on in the previous paper, correct? Not just the same methods?
184: a clear phrasing would be “a constant temperature of 24C and a constant humidity of 69%” or something similar.
*191: By “surface” do you mean area or perimeter? I assume the former, but you should use that more precise word instead. That would make this metric the relative size of the wing tip.
196-197: If eyespot size was bimodally distributed, why bother with the square root transformation? It still won’t be normally distributed. (Although, I guess this could make the residuals normally distributed).
*197-199: This shorthand is fine when conducting the statistical analyses and discussing them in the text, but is this also what you plotted? In this case, you need to explicitly note in the relevant axes that what you are plotting is the square root of the phenotype in question, on both the axis and in the figure caption. My preference would be to back-transform everything to the un-transformed axis (it’s closer to its biological interpretation), but the transformed plot is fine so long as you are clear that is what you are doing. (And because people will look at just the figures, it needs to be clear there, not just in the methods).
201: While it’s okay to test the sexes separately, if you make the revisions to test the treatments together, it would be best to also test the sexes together and add that as an additional factor.
*207: Ordering the plants by effect makes it almost impossible to visually compare figure 3c and figure 4c (for example, to see if the same plants have similar effects on the two traits). You should use a single, shared ordering of the plants across all figures.

Results:
255-257: This is a very telling result, and I think some of the best evidence you have for plasticity independent from growth rate. It comes across in figure 5 as well but it would be even more clear if you added the figure showing how growth rate changes with treatment to compare with figures 3/4.

Discussion:
287: Because your sample sizes are just in the tables, it would be good to give total sample sizes for each experiment in the methods or results of that experiment. As is, this is the first place you report an overall sample size.
*301: this pattern of wet-season forms on host-plants with faster growth is not at all easy to see in your results as is because it is not clear how host plant affects growth rate. Adding this relationship as a direct plot and making the order of host plants consistent across plots (maybe ordering them based on growth rate in figures 3 and 4 too) would make the evidence for this statement easier to see. This is especially needed because you are not using any overall statistical tests to determine the magnitude of this relationship (or its statistical significance).
340: If you want to provide more context for the separate regulation of these traits, you could simply measure the correlation between these traits or plot their relationship. Either could give you some idea of how strong this independent regulation is.

Conclusions:
362: I do think the conclusions would add much more to the paper if you added a couple more sentences putting these results back in the broader context of your introduction. It’s fine as is, but does not add much to the paper.

Table 1:
The caption would be easier to read if you put the definition of your parameters (e.g. S.E., Statistic, etc.) before the definition of the species.

Figures:
Figure 1: I really like this depiction of your hypotheses! It is nice and clear.
*Figure 2: as noted above, pleas fix the color key in this caption and ideally formally define what landmarks you used.

Appendix 1:
*If there are any method differences here from the temperature experiment besides the use of only 3 treatments, please explain them here. If they truly were the same except for the different treatment set, please state that here (rather than just the ambiguous “similar”).
Since it’s an appendix with plenty of space, you could optionally reproduce part a) of Figures 3 and 4 here on the same axis so the reader can directly compare the results. If not, I would recommend using specifically the same y-axis scale as figures 3a and 4a so readers can compare them side by side.
*I assume you mean the results are “qualitatively similar” not just similar since there is no evidence of a quantitative comparison. Please state this. Also cite figure 4 in addition to two.

Reviewer 2 ·

Basic reporting

OK

Experimental design

The design is OK but not optimal and some parts appear to have failed for technical reasons but these shortcomings, most likely, are not serious enough to affect main conclusions. The authors do not hide the problems which is nice.

Validity of the findings

Statistical treatment has to be improved, see below. It is however unlikely that the improved statistical treatment would lead to qualitatively different conclusions,

Additional comments

This is not a major contribution to science but definitely not a useless piece of research either. The value of the results is discounted by some methodological and technical problems which nevertheless do not appear fatal with respect to at least some conclusions drawn from this study. The authors nicely discuss the problems and limitations posed by them, and by doing so, do not attempt to oversell their study. This is laudable. A further unclear aspect of the methodology is how were the offspring of different females distributed between the treatments – might the treatment effects be confounded by genetic differences? This has to be discussed.

A major problem is however that statistical treatment appears incomplete and requires more attention. There appear to be no tests for among-treatment differences in the response variables?! The same appears to be true for the differences among sexes: the readers are simply instructed to infer the differences from the images. The within-treatment relationships between growth rate and the focal traits are analysed separately for each treatment (Tables 1 and 2) which results in low statistical power of these analyses due to low sample sizes. You should combine all the data in one united data set to analyse them jointly. As far as I see, there are two ways how this could be done 1) you can use nested ANOVA and test for the effect of growth rate so that the effect of growth rate is nested into the effect of treatment; 2) you can standardize the within-treatment distributions so that the averages of growth rate do not differ among the treatments (just by subtracting the treatment means, as the simplest way). Also, in accordance with what do you like to conclude from the data, you should perhaps test for the differences in the among- and within-treatment effects of growth rates on the focal traits (are the within- and among-treatment slopes different?); there are ways to do this, consult a statistician. Also, on the contrary to what you say, it cannot be easily seen from Figure 3 that the distributions are bimodal, perhaps you need a separate graph showing this directly. Also, if the distributions WERE bimodal, this would mean a problem for you because assumptions of regression analysis would be violated. The latter need attention in any case.

I have two comments on the interpretation of the results. First, part of the within-individual differences in growth rate must be genetic. It may well be the case that the larvae respond differently to genetic and environmentally based differences in growth rates which can partly explain the differences in within- and among treatment relationships. I would discuss this. Second, I think that the relationship between growth rate and wing shape may well be based on the relationship between body size and wing shape (so that growth rates are not causally involved): small and large butterflies may differ in shape because of some underlying ontogenetic processes/constraints or also in an adaptive fashion: for aerodynamic (or maybe also other) reasons, optimal wing shape may be different for differently sized individuals. You could try to separate the effects of growth rate and body size but this may not be easy because of the likely strong correlation between these traits. In any case, I strongly suggest that you discuss this.

Other points (mostly minor):

Line 29 (and elsewhere): it may be too ambitious to say that plasticity MAXIMIZES fitness
Line 32-33: dry season forms expressed in dry season: redundant
Line 56. I would not say “species” here
++ Is this the first experimental study of seasonal polyphenism in M. leda? Or perhaps even the first for any tropical satyrine apart from Bicyclus? If yes, I would advertise this more loudly and express happiness about this fact. If not, it must be said what is similar and what is different from previous studies.
++Any adaptive explanation proposed for the seasonal wing shape difference in Bicyclus?
Lines 58 and 74. The two “cascade”-sentences sound disturbingly similar, please reformulate.
Line 95. led -> leda
Line 96. Perhaps it is nowadays possible to replace the “distantly” by a numerical estimate of divergence time. Also, how common is seasonal polyphenism in such traits in the larger clade including both Bicyclus and Melanitis? Just these two genera or others as well?
Figure 2. Something is wrong here, I do not see a white triangle. Also, I would add a bit more explanations here, “yellow triangle as a proxy of wing shape” remains obscure here.
Line 181. “fourth batch” remains unclear here.
Tables 1 and 2. You have not mentioned the independent variable.
Line 328. Measurement error may also contribute low R2 values. Any comments?

---

## Round 0.2 · Major Revisions

Thanks for submitting your work to PeerJ. Please address these changes then submit.

Reviewer 1 ·

Basic reporting

As in its first submission, this manuscript is quite well written and structured; there have only been improvements in this regard, particularly to the figure captions.
The clarity of the provided raw data and code have been improved, but still have a few small details to further improve.
For the raw data, the included readme is beneficial, but it should also include units for the measurements where applicable (even if the unit for area is an arbitrary one, like pixels, that should be clarified). It would also be good for the readme to start with a brief description of the purpose of the file, so the two files can more easily be distinguished, since file names (while useful) may not always be clear.
The new combined code file is well organized, but I noticed one notable problem: it refers to a data file (Mleda_PP_ForewingAveraged_Cleaned.csv) that is not included with the upload. It could be that this is the file is another name for LedaPPaveragesNEW, in which case the name should be fixed. Otherwise this other file needs to be provided.

Experimental design

The clarity of the experimental methods has been satisfactorily improved by the additional details provided in the revisions.
The statistical approach has been majorly improved, and the study greatly benefits from this, but how the statistics are reported needs more detail. The current description (lines 211-216) needs more detail on the specific models that were tested. Some of the these details can be found in the header for table 1, but others I have had to infer through detailed reading. Both should be explicitly stated in the methods section. Most importantly, the authors should state the structure of the full model tested and used for model selection (fixed effects, interactions, and mixed effects). They should also move most of the information from lines 1-5 of the table one heading to the methods, such as the method used for conducting statistical tests (the table heading can then include a shortened version of this).

Validity of the findings

Most of the authors’ interpretation and discussion of their results remains appropriate given their new statistical analyses; however, there is now one claim that I think merits reconsideration given the final results.
Specifically, that temperature has an effect on seasonal phenotype independent of growth rate is clear from both figures and the statistical analysis; however, it is not so clear to me whether host plant has an effect independent of growth rate. This doesn’t undercut the authors’ final conclusion (growth rate can’t be an integrating factor since that would require it to mediate the effect of both treatments), but the authors should either a) acknowledge the uncertainty regarding the pathway from host plant to growth rate to seasonal phenotype in their discussion or b) find a way to strengthen this argument (show that the effect is weak).
As is, I think both the graphical and statistical arguments that growth rate couldn’t mediate the connection between host plant and phenotype are weak. To make this argument you need to essentially demonstrate that the effect of the treatment is stronger (or otherwise different) from the effect of growth rate within treatment. The current argument appears to me to mainly rely on the existence of some outliers in the slopes of individual treatments, but given the large number of individual treatments and sometimes small sample sizes per treatment, this could easily be caused by chance.
Graphically, in figure 8c and especially figure 9c, while there is some variation in the reaction norms on individual plants (the scatter cited in lines 276-277 may in part likely be caused by noise and small sample sizes on some plants), there is general alignment in the direction of many of the reaction norms and that direction doesn’t clearly visibly differ from the overall trend across treatments (although this is difficult to visualize given the strong overlap among treatments). This is in contrast to the temperature experiment (figures 8a and 9a) where the trend within treatments, although consistent, is visibly much weaker than the trend across treatments. If you want to strengthen the graphical argument, you would need to find a way to more clearly visualize this difference between the within and across treatment trends (for example, maybe drawing the across treatment lines).
Statistically, you do find a highly significant effect of growth rate whereas support of an effect of treatment independent from growth rate is mixed (plant is not included in the selected model). This doesn’t prove that growth rate mediates the effect of host plant, but it also can’t be used to dismiss this possibility either. Again, this contrasts with the temperature experiment where although there is also a significant within-treatment effect of growth rate on phenotype, there is also a clear independent effect of temperature treatment even after accounting for growth rate. I’m not certain of a way to statistically compare the effects of growth rate within and across treatments (an ideal test) for your data, and I don’t think that is essential, but unless you can show the slopes of these two lines differ, you cannot conclude that growth rate can’t explain the effect of plant treatment. If you want to make this argument, one detail that might help would be to mention the actual effect of growth rate in the results text (currently it’s only in the supplemental tables such as A2.3) and discuss how while it’s significant it is to low to explain your results (if this argument can reasonably be made). This could be done more effectively if you also find a way to estimate the across treatment slope relating growth rate to seasonal phenotype.
None of these adjustments to your statistics or graphs are required, but if you keep your analysis as is, you should acknowledge the uncertainty of whether growth rate can explain the effect of host plant in your discussion. Again, this doesn’t cause a problem with the main conclusion—because there is clear evidence that growth rate can’t explain the effect of temperature, it can’t be an integrating factor—but it does change the nature of the evidence that supports that conclusion.

Additional comments

Line by line comments. These are mostly small suggestions for improvements. Those that I think are necessary to address are indicated by *:
38: Maybe you should specify “external cues” since you earlier refer to internal cues as well.
167: the combination of two separate statements in the parenthetical is a little confusing and makes it hard to tell what the third generation is referring to.
211-213: these are the lines that need more detail regarding the statistical analysis.
213-216: I appreciate the detail you provide on the next lines regarding the packages used, although you should ideally also provide the version used, especially for R.
254: It’s not essential but it might be interesting to note that the range of eyespot sizes produced by host plant is notably smaller than that produced by temperature (comparing the scales of Figure 4a/b and 4c)
294-296: this is the claim I think needs better support in my criticism above. I am not convinced that the relationship between larval growth rate and phenotype is weak specifically for the diet experiment (here you discuss both together), more so for wing shape (which you do acknowledge). Providing some quantitative values here could help strengthen your argument.
310-311: This argument regarding high scatter among treatments is from the previous draft of the manuscript, and as then, I still do not find it a compelling argument since at least in some cases that extreme scatter could easily be caused by small within treatment sample sizes.
323-325: I am not clear on what you mean by an “L-shaped reaction norm” here and elsewhere since your reaction norms in the plots at least to me seems to be a much simpler linear negative relationship. Could you please clarify what you mean in this case or alter your phrasing?
*Figure 3: please provide a unit for growth rate in you Y axis. Also, in your caption you state that No true dry-season… so the y-axis was truncated”. This figure is growth rate, so is this just a copying error from the other figure captions?
*A2.4, A2.8: should read per host plant
*A3.1 should reference A3 not A2, and there are more similar mistakes. Please double check these references overall.

Reviewer 2 ·

Basic reporting

OK

Experimental design

OK

Validity of the findings

Some attention to statistical analyses is still needed. Explained below.

Additional comments

No doubt, the authors have invested lots of effort in improving their manuscript which is laudable. Nevertheless, I still see two problems with statistical treatment which must be addressed.
First, in the analyses in Table 1 (and their counterparts in Supplementary material) there are interactions between categorical and continuous variables included. To the best of my understanding, this is problematic because in such models, the main effects of categorical variables are tested assuming that the value of the continuous variable equals zero which is frequently not what we want to know. Please tell me if I am wrong. To tackle the problem, you should standardize the continuous variables in such a way that their sample means equal zero (just subtract the mean value) (it is not stated that this was done) in which cases the main effects of the categorical variables will be analysed assuming that the continuous ones have their mean values in the sample (which usually makes much more sense).

Second, to separate the direct effect of growth rate from simple allometric relationships you should SIMULTANEOUSLY include growth rate and an index of body size (either pupal mass or wing size) into your glm models (may be enough if you do this for your Selected models in Table 1, curious to see what happens!). If the effect of growth rate will disappear or will be substantially weakened, you should conclude that there is not much evidence supporting a causal effect of growth rate, and everything can be explained by allometry. My feeling that such an analysis is a must if you wish to keep the focus of the paper on possible causal effects of growth rate as a cue.

Otherwise, the Results section of Abstract sounds like studying the effect of growth rate as cue was the only goal of the paper. Actually, there is a LOT of other valuable information about these reaction norms which the Abstract should proudly present.

Minor
Line 57. I would not talk about butterflies in this paragraph, try to be more general and switch to butterflies in the following paragraph.
Line 96. When mentioning the study species for the first time, I would clearly say that this a tropical species.
Line 132. I would have the paragraph break before “In most cases…”, and I would say “the PARENTAL butterflies”. I would then omit paragraph break at line 136, and would delete “In summary”.
Line 137. How many larvae per sleeve.
Lines 142-147. This information appears largely irrelevant in the context of this paper and may be deleted.
Line 207. The story about the pilot experiment remains little obscure here.
Line 210. How the plants were ordered belongs to Figure text?
Line 211. Clearly, you have to tell more here about the analysis in Table 1.
Line 318. The temperature-size-rule is a very general phenomenon applying to most ectotherms so that such case-specific adaptive explanations may not fly high.
Lines 326-336. You are telling quite trivial things here, and beyond trivialities, you have no information about the origin of these within-treatment differences. You may just say that they were genetic and/or resulting from differences in the microenvironments, or something like that.

---

## Round 0.3 · Minor Revisions

Thank you for your submission to PeerJ.

Please address these final changes and resubmit.

Reviewer 1 ·

Basic reporting

Good

Experimental design

Good

Validity of the findings

Good

Additional comments

I am happy with all of the changes made in the latest revisions to both the main manuscript and the data files and have no further requests.

Reviewer 2 ·

Basic reporting

OK

Experimental design

OK

Validity of the findings

A comment on statistics not satisfactorily considered, see below.

Additional comments

Dear Authors,

Thank you for your efforts but, I am sorry, I am stubborn.

Last time I wrote:

“Second, to separate the direct effect of growth rate from simple allometric relationships you should SIMULTANEOUSLY include growth rate and an index of body size (either pupal mass or wing size) into your glm models (may be enough if you do this for your Selected models in Table 1, curious to see what happens!). If the effect of growth rate will disappear or will be substantially weakened, you should conclude that there is not much evidence supporting a causal effect of growth rate, and everything can be explained by allometry. My feeling that such an analysis is a must if you wish to keep the focus of the paper on possible causal effects of growth rate as a cue.”

… so that I would expect that 1) either you do such an analysis or 2) you provide a straightforward explanation for why you prefer not to do so. I cannot find either. There is lots of (too much?) stuff in Appendix 3 (but no mention of it in Results sections Growth rate influencing eyespot size and Growth rate influencing wing shape!) but I could not find what I was looking for: a GLM table with both growth rate and wing size simultaneously included as predictors (for both eyespot size and wing shape as response variables). I would like to see if growth rate would remain significant or not. Please without any model selection procedures or similar, just add „wing size“ to table 1 in an alternative set of analyses, please. Of course, here is high collinearity between growth rate and wing size but this is not a problem as long as you ask the question I have specified above.

If growth rate will not remain significant, this will substantially strengthen your message of no causal effect of growth rate: everything can be explained by body size.

Otherwise, just one suggestion: in the Results section of Abstract, please add one general descriptive sentence before reporting the results on the causal effect of growth rate, perhaps one like you have in the Conclusions: reaction norms were similar to those in B. anynana. I still think that you have lots to report and advertise also at the descriptive level!

---

## Round 0.4 · accepted · Accept

Congratulations.

Thanks for submitting your work to PeerJ.